# Through the Looking Glass: Unraveling the Stage-Shift of Acute Rejection in Renal Allografts

**DOI:** 10.3390/jcm11040910

**Published:** 2022-02-09

**Authors:** Reuben D. Sarwal, Wanzin Yazar, Nicholas Titzler, Jeremy Wong, Chih-hung Lai, Christopher Chin, Danielle Krieger, Jeff Stoll, Francisco Dias Lourenco, Minnie M. Sarwal, Srinka Ghosh

**Affiliations:** 1NephroSant Inc., 1900 Alameda de las Pulgas, San Mateo, CA 94403, USA; rsarwal@nephrosant.com (R.D.S.); WYazar@nephrosant.com (W.Y.); NTitzler@nephrosant.com (N.T.); jwong@nephrosant.com (J.W.); clai@nephrosant.com (C.-h.L.); cchin@nephrosant.com (C.C.); DKrieger@nephrosant.com (D.K.); jstoll@nephrosant.com (J.S.); Francisco@nephrosant.com (F.D.L.); 2Department of Surgery, University of California, 400 Parnassus Ave, San Francisco, CA 94143, USA

**Keywords:** renal transplant, acute rejection, urine, cell-free DNA, multi-analyte, biomarkers

## Abstract

Sub-optimal sensitivity and specificity in current allograft monitoring methodologies underscore the need for more accurate and reflexive immunosurveillance to uncover the flux in alloimmunity between allograft health and the onset and progression of rejection. QSant—a urine based multi-analyte diagnostic test—was developed to profile renal transplant health and prognosticate injury, risk of evolution, and resolution of acute rejection. Q-Score—the composite score, across measurements of DNA, protein and metabolic biomarkers in the QSant assay—enables this risk prognostication. The domain of immune quiescence—below a Q-Score threshold of 32—is well established, based on published AUC of 98% for QSant. However, the trajectory of rejection is variable, given that causality is multi-factorial. Injury and subtypes of rejection are captured by the progression of Q-Score. This publication explores the clinical utility of QSant across the alloimmunity gradient of 32–100 for the early diagnosis of allograft injury and rejection.

## 1. Introduction

It is not surprising that advances in immunosuppression treatment for kidney transplant recipients have not had a more dramatic impact on the improvement of long-term allograft survival [1]. A pivotal reason for this is the inability to facilitate a stage-shift rejection surveillance model that will detect early the molecular and cellular transition of the graft from health/ immune quiescence to acute rejection. There is an early state of injury with minimal impact on the allograft for development of progressive fibrosis and tubular atrophy [2]. Recent studies [3,4] have underscored the temporal lag between the onset of intra-graft molecular signals of rejection [5]—where the allograft has patchy histological injury, frequently missed by a needle biopsy (Figure 1) with putative function [6]—and severe dysfunction.

There is a critical unmet need to uncover this early stage-shift of acute rejection when injury is mild and potentially reversible. This requires the diagnostic to independently detect rejection, without the testing trigger being a change in graft function [7,8]; also be able to risk stratify the severity of rejection without the inherent reader bias observed for the allograft biopsy [9]. The clinical validity of a diagnostic with this specification will provide an improved standard for allograft surveillance, enabling real-time immunosuppression titration.

A non-invasive, urine biomarker based multi-analyte diagnostic test, QSant was developed and clinically validated [10]—to address this unmet need. QSant provides a composite Q-Score, scaled between 0 and 100, to quantify risk of alloimmune injury. The Q-Score has the unique capability to expose a multi-dimensional view into the allograft. Q-Score <32 captures a stable, immunosuppressed allograft and a Q-Score >32 detects the changing trajectory of injury in allograft rejection. As the Q-Score progresses from 55 to 100, there is increased concordance with high-grade histological rejection, as assessed by Banff [8].

The intent of the initial [10] Q-Score development study (*n* = 364) was to dichotomize an immunosuppressed renal transplant population into a two-class model—stable (STA) and acute rejection (AR)—with two independent validation cohorts (*n* = 162) to prognosticate rejection risk. Additionally, there was an independent prediction cohort (*n* = 91) to assess Q-Scores in borderline acute rejection (bAR), ad-mixed cases of AR and chronic allograft injury (tubular atrophy and interstitial fibrosis) and BK viral nephritis with and without the presence of tubulitis.

This study demonstrates the clinical utility of serial monitoring via QSant in prospectively collected longitudinal patient urine samples. A review of a large dataset of contemporaneous kidney allograft biopsy and QSant data, inclusive of real-world data (RWD) from 11 US kidney transplant centers, show that the Q-Score offers a dynamic view into the changing state of allograft injury and rejection in response to alterations in the patients’ immunosuppression regimen. Early detection of rejection, specifically in the 32–55 range, when graft function and biopsy may yet only be minimally perturbed, promulgates early and personalized care pathways for allograft rejection. Thus, precision monitoring with the Q-Score has the potential to not only improve a transplant recipient’s’ quality of life but to also alleviate the financial burden on national health care budgets, by minimizing unnecessary biopsies, untimely transplant loss, incidence of chronic dialysis, and overall morbidity.

## 2. Methods

The following two-pronged analyses were conducted to elucidate the performance characteristics of QSant in the context of a spectrum of alloimmune injury and to observe the Q-Score distribution in our cross-sectional retrospective data and compare this with a real-world data paradigm. First analysis included Q-Score distributions in the biopsy-paired validation (*n* = 162) and prediction (*n* = 91) data from the Yang cohort, which were collected between 2010 and 2018 [10]. Second was a new analysis of a real-world data (RWD) cohort, comprising of prospectively collected urine samples (*n* = 235) from 11 different US transplant centers, collected between 1 April and 1 October 2021. All RWD fresh urine samples were self-collected by patients in urine cups and then transferred to vacuum tubes, which were pre-filled with proprietary preservative buffers. These buffers protect against sample degradation during shipment to the laboratory for testing. Samples were processed in the NephroSant CLIA Laboratory in Brisbane, CA, USA. The following biomarkers were analyzed: protein: clusterin, total protein, the inflammation marker CXCL10; metabolite: creatinine and DNA: the amount of cell-free DNA (cfDNA), fraction of methylated cfDNA(m-cfDNA). FDA-approved tests on the Beckman Coulter AU400 analyzer were used for the measurements of urine creatinine and urine total protein, and proprietary, optimized ELISA-based assays were used for the measurements of cfDNA, m-cfDNA, clusterin, and CXCL10. All QSant assays were performed on patient urine samples collected prior to treatment intensification of AR.

In the Yang cohort, AR samples comprised of a spectrum from early and mild acute rejection, Banff defined borderline rejection [11], histological rejection without graft dysfunction (normal graft function by a serum creatinine test; subclinical AR or sAR), and histological rejection with Banff grades [11] from grade I to II and with either antibody-mediated AR (ABMR) or T cell-mediated AR (TCMR). Patients received maintenance immunosuppression with tacrolimus, mycophenolate mofetil, and steroids with T cell depletion induction or IL2R monoclonal antibody. Clinical annotation for biopsy reasons such as for-cause or protocol were available for these samples. For the RWD cohort, patient biopsy was not mandated but was disseminated at investigators’ discretion. Patients spanning both pediatric and adult age groups were on differential immunosuppression for maintenance (tacrolimus, mycophenolate mofetil. everolimus, sirolimus, tocilucimab, prednisone) and induction (thymoglobulin, IL2R monoclonal antibody, belatacept).

## 3. Results

The RWD cohort (*n* = 235) of both adult and pediatric renal transplant patients demonstrated a spectrum across the Q-Score distribution (Figure 2). The observed median Q-Score was 37. Specifically, 37.9% of the samples were enriched for immune quiescence (Q-Score < 32); the remaining 62.1% were enriched for the injury/rejection spectrum (Q-Score ≥ 32). The Anderson–Darling Test of goodness of fit rejected the null hypothesis that the Q-Score was gamma distributed (*p*-value: 0.009801); a Gaussian mixture-model fit to the distribution identified primary partitions approximating Q-Scores of 32 and 55 (Figure 3). Of the 62.1%, 49.8% enriched the autoimmune flux domain of Q-Score 32–55 and 12.3% enriched the higher-grade AR domain of a Q-Score ≥ 55. The chronicled cohort had allografts ranging from 6 days to 27 years post-transplant and therefore presented a realistic sampling of the burden of rejection across a transplant patients’ journey [12]. Stegall et al. highlight the differential prevalence rate of TCMR versus ABMR rejection in the periods 6 months to after a year post transplant. This is superimposed by an undercurrent of borderline rejection and other chronic graft injuries. Thus, we posit that in the 32–100 range, there is an elasticity in the expression of alloimmune injury severity in AR. Therefore, it is more apt to consider AR as a spectrum—encompassing a mixed population of rejection subtypes—early, mild, mixed rejection, and pure TCMR and ABMR.

A binary classification of STA and AR has been proposed by many diagnostic tests [13,14]. QSant has a multi-analyte composition that results in a scaled Q-Score that directly correlates with the amount of tubulitis and inflammation [10] in the allograft undergoing acute rejection. Due to the unique performance characteristics of QSant, we propose the following model (Figure 4) to refine the more simplistic binary classification for rejection and no rejection.
(A).*The Immune quiescence domain of Q-Score < 32*: Allograft health is demarcated by this domain; this has been validated in published data [10] and classifies stable (STA) allograft recipients. Patients in this range are on immunosuppression that is adequate to suppress alloimmunity and persisting low Q-Score < 32 may possibly allow for immunosuppression to be safely maintained or diminished.(B).*Alloimmune injury domain of Q-Scores in the range of 32–55*: expose a state of alloimmunity flux as the graft switches from a state of allograft health to early alloimmune injury. This domain enriches for molecular allograft injury, where early rejection may not have histologically developed fully yet and could be missed by a biopsy [15].(C).*High-grade acute rejection domain of Q-Scores in the range of 55–100*: provide a zone of advanced alloimmunity, associating with simultaneous histological and molecular rejection injury.


Serial monitoring by QSant across a subset of 32 patients (RWD cohort,) over a period of two consecutive quarters (timepoints 1 and 2), captures the essence of the alloimmunity flux (*p*-value: 0.086 by non-parametric Wilcoxon test) (Figure 5). It is worthwhile highlighting that this is not a powered randomized control trial but rather a real-world evidence-driven study. To ensure a uniform sampling frequency for the RWD data obtained between 1 April and 1 October 2021, only two serial tests could be included. The analysis reflects an alloimmunity flux trending toward statistical significance. Patients in the immunoquiescent domain (31% [10/32]) by and large continue to be stable (with Q-Scores that track at <32) 22% [7/32]). However, 9.4% (3/32) of patients show Q-Score drifts above 32 with evidence of early rejection. Patients in the alloimmune injury domain (53% [17/32]) follow three primary trajectories:
(i)A planar gradient in Q-Score across time, showing persistence of alloimmunity (28% [9/32]);(ii)An increasing gradient, delineating patients moving into the mature AR domain (3% [1/32]);(iii)A decreasing gradient, delineating patients moving into the immunoquiescent domain, putatively in response to immunomodulation following assessment of chronic/active/mixed rejection (22% [7/32]). Albeit a small sample size (16% [5/32]), the evolving and resolving nature of higher-grade acute rejection (Q-Score > 55) in response to immunomodulation is exemplified here.


The observed alloimmunity flux lends to the suggested QSant monitoring protocol (Figure 6) with a differential approach in response to the evolution or resolution of injury/rejection. The ΔQ score change of 10 is commensurate with the 25% increase in the serum creatinine, which is the standard of care for renal transplant management.

Following the observations in the RWD study, a re-analysis was performed on the Yang cohort, and performance was characterized across the combined validation sets (*n* = 162). For a dichotomized cohort of STA versus AR, the cut-point of 32 was recapitulated (*p*-value: 2.26 × 10^−29^); specificity and sensitivity were established at 94% and 99%, respectively, with an AUC of 98% (95% CI: 0.98 to 0.99, *p* < 0.0001) (Figure 7 and Figure 8). To address the class imbalance (STA:AR = 98:64), the performance was validated with a geometric mean of 96.5%. The inflexion from stable renal allograft to early acute rejection related to molecular inflammation, as identified by increasing Q score (>32), classifies many AR samples with additional mixed phenotypes (chronic injury, BK viral nephritis) or biopsy-confirmed phenotypes of bAR with tubulitis. The re-analysis with the inclusion of the bAR population with a hyperparameter optimized Q-Score model confirmed a cut-point of 55 with an AUC of 83% and geometric mean of 82% (Figure 9 and Figure 10), demarcating the higher-grade AR. Again, given the class imbalance, a measurement of geometric mean is most relevant. Analysis of clinical metadata in the alloimmune injury domain further revealed an enrichment for sub-clinical acute rejection (sAR)—rejection in the absence of graft dysfunction. Since there is no functional decline of the allograft, elevation of serum creatinine—the Standard of Care—is not implicated here. The current detection mechanism for sAR is a positive result from protocol or routine periodic biopsies done without any established clinical cause. For the Yang cohort, the intermediate zone (Q-Score: 32–55) encompassed a near equal admix of protocol (*n* = 39; 43.3%) and for-cause (*n* = 51; 56.7%) biopsies. The protocol biopsies are enriched for sub-clinical rejection, 95% (37/39) whereas for cause biopsies enrich for AR at 54.9% (28/51). Published biopsy correlation data [10] show that samples in the higher-grade AR domain (Q-Score > 55) had a statistically significant correlation with inflammation (I-score; *p* < 0.0001) and tubulitis (t-score; *p* < 0.0001) scores.

## 4. Discussion

The etiology behind alloimmune rejection in renal transplantation is complex. This is further complicated by the differential health of the renal parenchyma of donor kidneys and the presence of comorbidities post-transplantation. The interplay of these factors lead to a dynamic spectrum of rejection inclusive of similar and disparate cellular and molecular mechanisms of ABMR and TCMR. The interplay of the individual component markers of QSant, that correlate with both the inflammation and tubulitis injury of acute rejection, highlights its clinical relevance for immunosuppression titration and serial of monitoring to evaluate the resolution of acute rejection. The quantitative nature of the Q-Score provides specific and detailed insights into early perturbations due to alloimmune injury in different compartments of the transplant kidney, which is not possible in other existing (blood) diagnostic assays, that only provide a single threshold for the presence or absence of acute rejection.

We highlight in this study that the evaluation of the health of an allograft has to be able to involve accurate assessment of the state of immune quiescence (Q-Score < 32), where immunosuppression minimization could be supported, as well as understand the phase-shift of alloimmune injury through early stages of molecular and cellular rejection where histological injury and clinical graft dysfunction are still lagging indicators [15] (Q score 32–55). Uncovering this early phase-shift in acute rejection provides an unprecedented window to proactively tweak immunosuppression and reverse allograft injury, such that the Q-Score declines to the quiescence range of <32. As discussed above, many clinically indicated and protocol biopsies may be unnecessary invasive procedures if the accompanying Q-Score in a paired urine sample is less than 32 [10]. Failure to recognize this early phase-shift puts us at the current clinical state of rejection diagnosis today, with clinical graft dysfunction and/or varying grades of histological rejection injury on biopsy (protocol or indicated), which are mostly associated with a Q-Score of >55. An additional benefit of the scaling of the Q-Score is appreciated here as the severity of inflammation (i-score) and tubulitis (t-score) on histology is correlated with the grading of the Q-Score from 55 to 100. Thus, the serial performance of QSant can optimize both patient and transplant clinical management.

## 5. Conclusions

In conclusion, the Q-Score offers the potential to be used as an immune-monitoring tool to guide the use of immunosuppression, with the ultimate goal of controlling subclinical intra-graft inflammation and thus prolonging graft survival. Further assessment revealed that the QSant has the potential to differentiate higher-grade AR episodes from borderline as well as sub-clinical rejection.

## Figures and Tables

**Figure 1 jcm-11-00910-f001:**
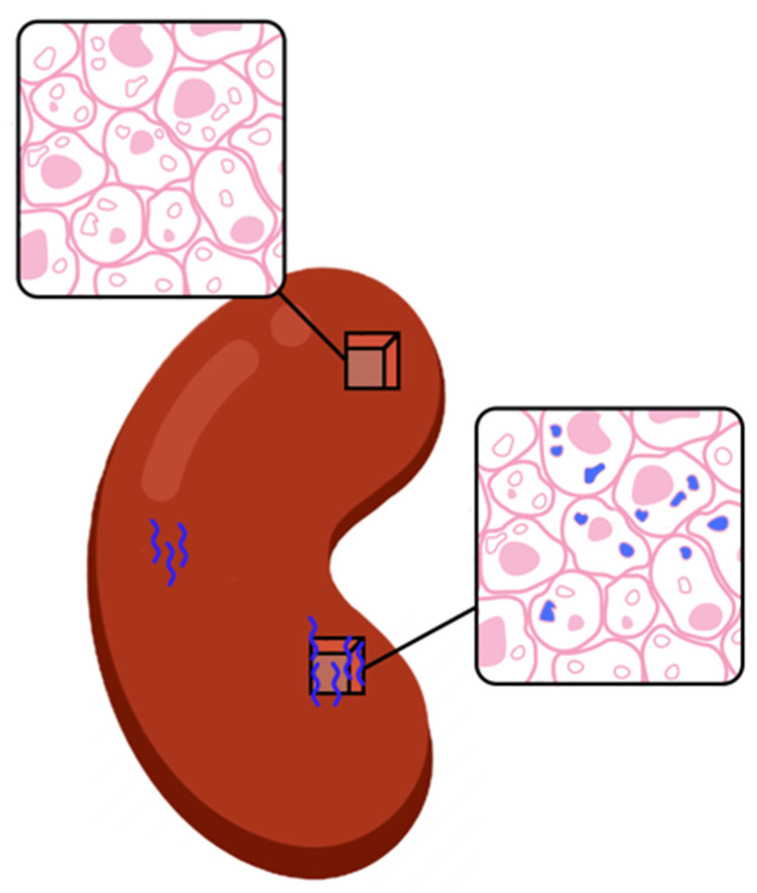
Early rejection is a histologically patchy process [4]. Renal allograft biopsies can return a false negative or true positive result based on the region of tissue sampling. The top *cube* samples tissue from a region of the kidney where rejection is not present and would result in a false negative. The bottom *cube* samples tissue from a region of the kidney manifesting rejection and would result in a true positive. Given the gold standard stature of biopsy, it would most definitely impact patient care. This is the inherent fallibility of the renal biopsy as the standard for rejection diagnosis.

**Figure 2 jcm-11-00910-f002:**
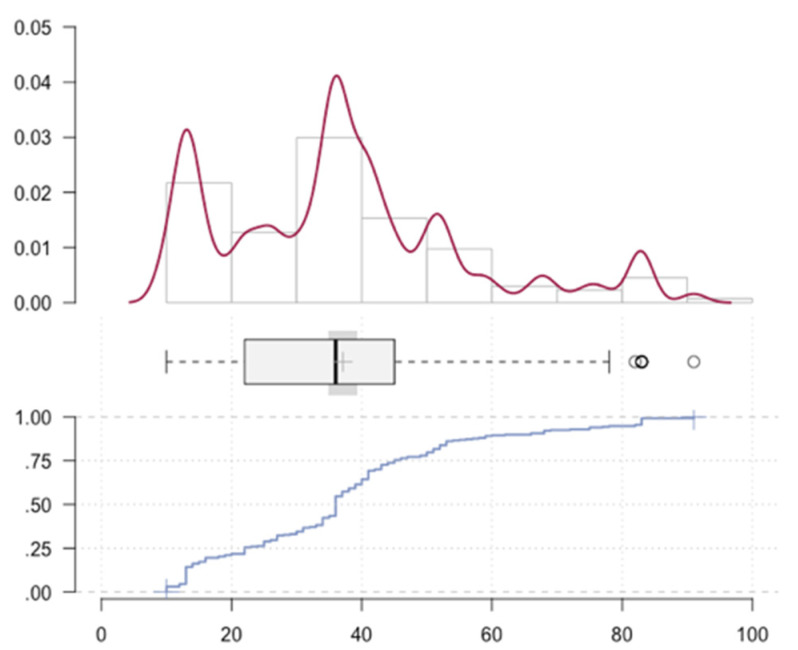
Q-Score distribution in the RWD cohort with a median at 37.

**Figure 3 jcm-11-00910-f003:**
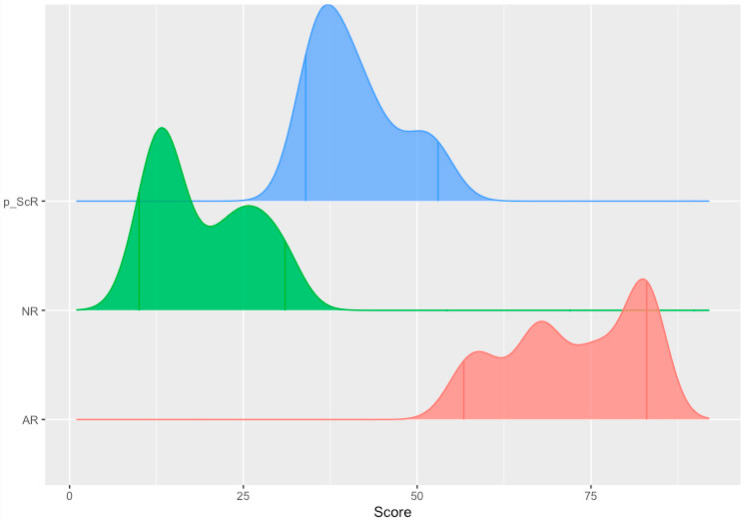
A Gaussian mixture model fit to the Q-Score distribution identifies partitions approximating 32 and 55. AR: Acute rejection, NR: no rejection, p_ScR: presumed sub-clinical rejection.

**Figure 4 jcm-11-00910-f004:**
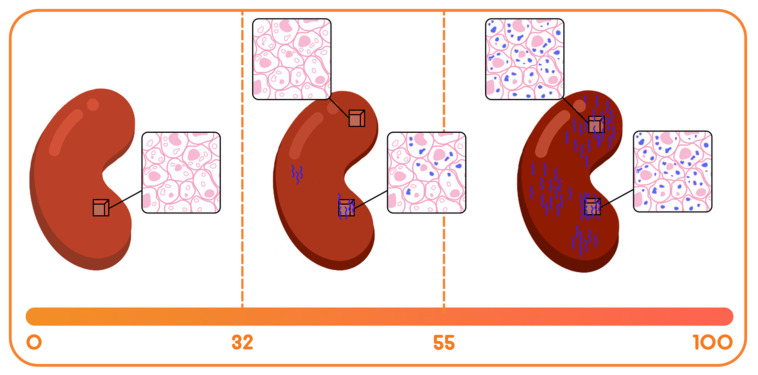
Increasing Q-Score reflects the progression of injury and underlying damage to the allograft. (**left**) Q-Score < 32 reflects a stable immune quiescent allograft status. Q-Scores in the 32–55 range (**middle**), as demonstrated here, have the capacity to detect early, sub-clinical, or borderline rejection, where the gold-standard biopsy can often miss histological signs of rejection depending on where the biopsy tissue sample was taken from. This is the alloimmune injury domain. Q-Scores > 55 (**right**) reflect a more mature or higher-grade acute rejection with histological and molecular alloimmune instability.

**Figure 5 jcm-11-00910-f005:**
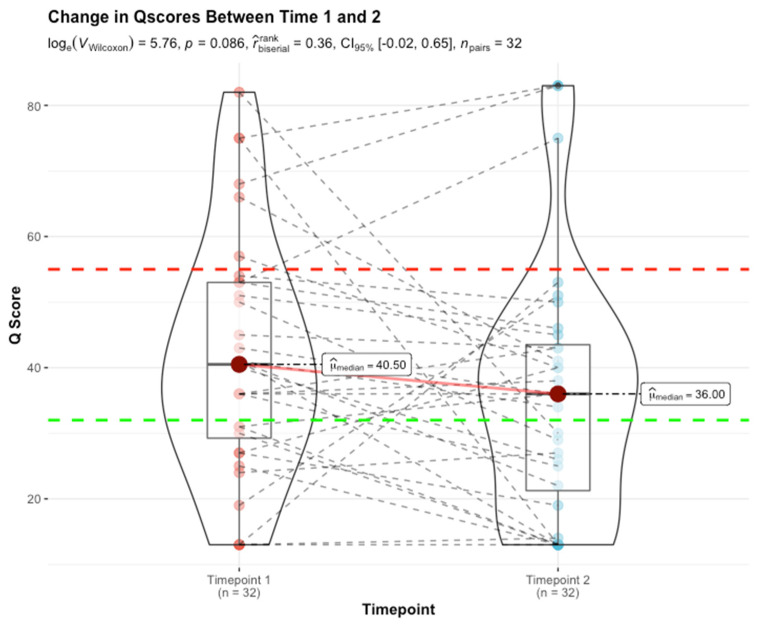
Trends analysis on serial Q-Score demonstrates the utility of QSant in monitoring allograft injury post-transplantation.

**Figure 6 jcm-11-00910-f006:**
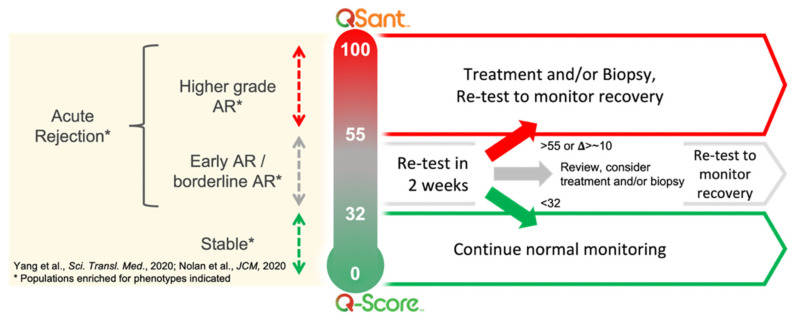
Potential QSant serial monitoring schedule for differential Q-Scores trend.

**Figure 7 jcm-11-00910-f007:**
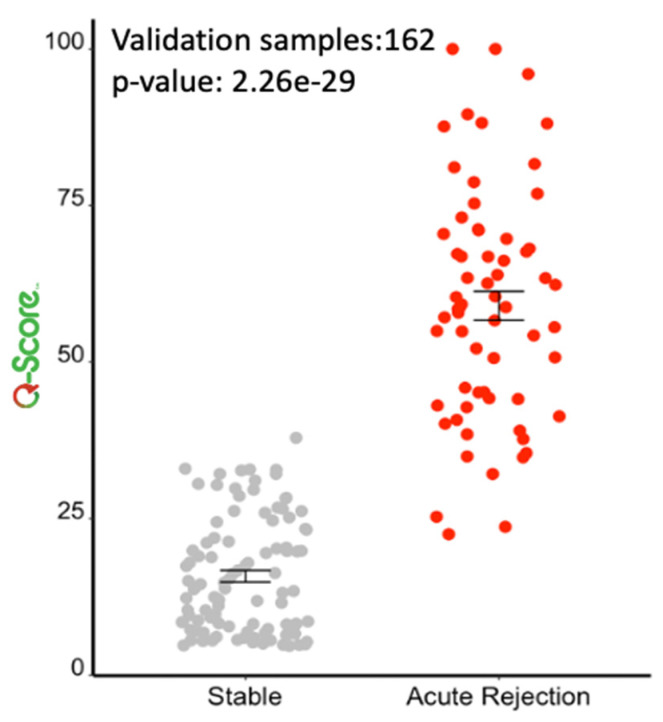
There is a statistically significant difference (*p*-value: 2.26 × 10^−29^) in Q-Score between STA and AR.

**Figure 8 jcm-11-00910-f008:**
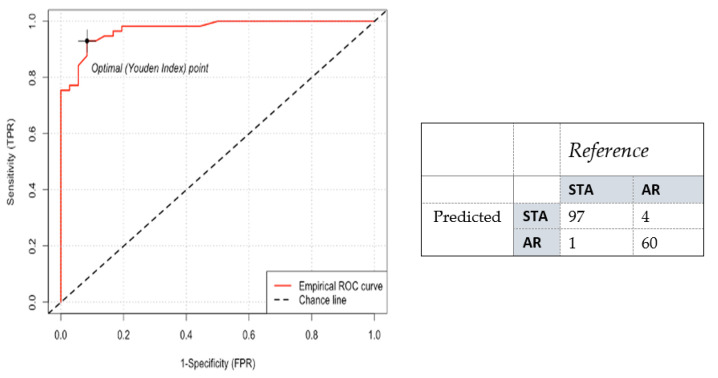
(**left**) ROC-AUC analysis of the Q-Score in the Yang validation sets shows excellent discrimination of AR phenotypes from STA, with an AUC of 98%; (**right**) The table represents the corresponding confusion matrix.

**Figure 9 jcm-11-00910-f009:**
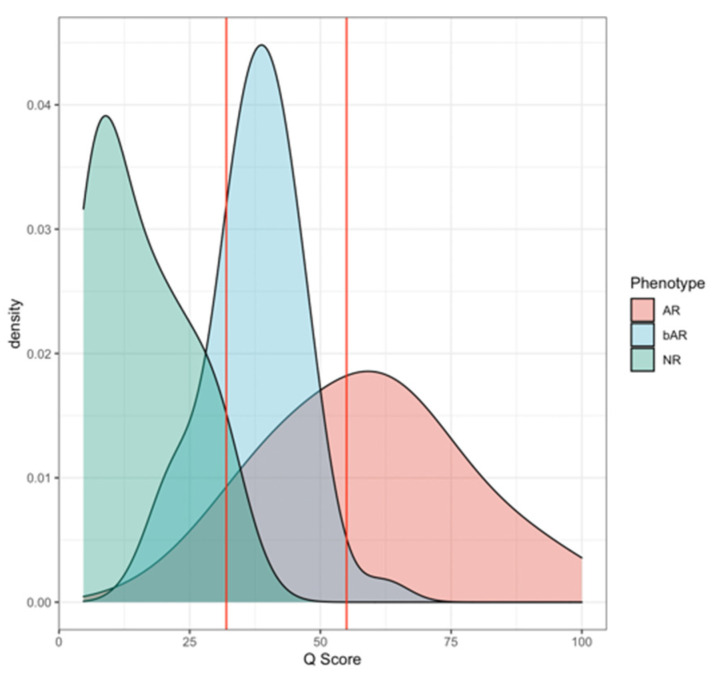
Distributions of Q-Scores in the Yang set reveal a three-mode population, with bAR phenotypes mostly having scores within the 32–55 range.

**Figure 10 jcm-11-00910-f010:**
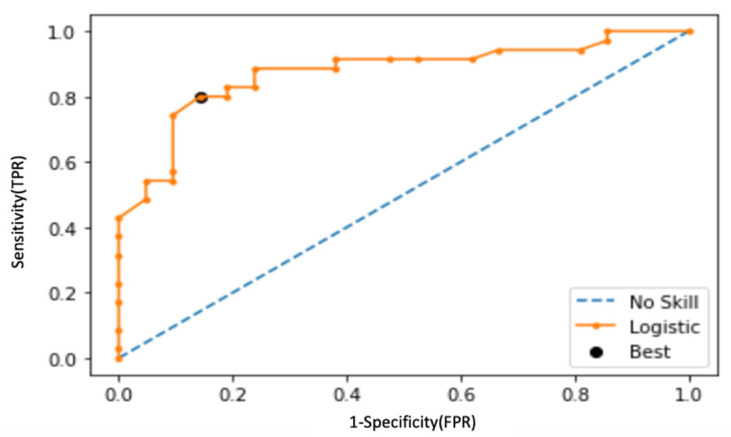
ROC-AUC analysis of Q-Score’s ability to pick up different rejection types reveals a G-mean of 83% at the cut-point of 55.

## Data Availability

Nephrosant had full access to the datasets in the study and full responsibility for the integrity of the data and accuracy of the data analyses.

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
