# Peer review of "Through the Looking Glass: Unraveling the Stage-Shift of Acute Rejection in Renal Allografts"

_jcm, 2022, doi:10.3390/jcm11040910_

Round 1

Reviewer 1 Report

The paper by Sarwal RD and co-authors explores the role and the clinical utility of QScore for the early diagnosis and effective management of allograft rejection.

The paper is interesting, updated and well written.

The paper does not need of substantial changes.  

Author Response

Thank you so much for reviewing the paper!

Reviewer 2 Report

Doctors caring for transplant patients are looking for an appropriate method of monitoring the allograft. The main cause of graft failure and loss is rejection. The presented work presents an interesting tool for such monitoring with clinical verification. Adequate immunosuppression, not too strong (risk of side effects) and not too weak (risk of rejection) is crucial for the long survival of the transplant and the patient. In my opinion, the publication presents a clinically interesting verification of a tool that can be used in the routine care of recipients. 

Author Response

(The authors gave the same response as above.)

Reviewer 3 Report

Authors present preliminary temporal data on a potential renal biomarker.

Concerns as follows:

1) Time series data in Fig.5 are two time points with 32 samples. Two time points and 32 samples seem insufficient

for actual data analysis, so that the clinical protocol recommendations in Figure 6 are premature.

2) Figures 8 & 10. State regression program, metrics, and misclassification table, confusion or error matrix.

3) Significant correlation between Q scores and Banff i score and Banff t scores is stated.  The significance of the correlation is meaningless if the slope/correlation is weak. State slope/correlation. i and t scores are ordinal variables not continuous as coded. Please explain. 

Author Response

Serial monitoring by QSant across a subset of 32 patients (RWD cohort,) over a period of two consecutive quarters (timepoints 1 and 2), capture the essence of the alloimmunity flux (p-value: 0.086 by non-parametric Wilcoxon test) (Figure 5). It is worthwhile highlighting that this is not a powered randomized control trial rather a real-world evidence driven study. To ensure a uniform sampling frequency for the RWD data obtained between April 1- October 1, 2021, only two serial tests could be included. The analysis reflects an alloimmunity flux trending towards statistical significance. 

Figure 7/8: Confusion matrix added as requested.

Figure 9/10: Performance characterization statistics re-stated in the context of geometric mean, given the class imbalance across the subtypes of rejection.

Regarding the biopsy data question: Published biopsy correlation data (10), show samples in the higher-grade AR domain (Q-Score >55) had a statistically significant correlation with inflammation (I-score; p<0.0001) and tubulitis (t-score; p < 0.0001) scores.

Round 2

Reviewer 3 Report

In Figures 8 & 10, it would be better to regress rejection types, not just binary, vs quant data.

This would inform readers about any ambiguity of the test will weak rejections.

Author Response

For figure 8: There are only 2 phenotypes: Stable(STA) and acute rejection(AR) with and without histological confirmation; therefore the dichotomization is the only approach.

For figure 9: It is correct that there are 3 phenotypes at play: AR with histological confirmation, borderline AR(bAR) and no rejection(NR) - the latter being equivalent with STA. However the ROC analysis in Figure 10 is discriminating between bAR and AR with histological confirmation;  therefore the dichotomization is the only approach.